# Trends and Emerging Hotspots in Toxicology of Chironomids: A Comprehensive Bibliometric Analysis

**DOI:** 10.3390/insects16060639

**Published:** 2025-06-17

**Authors:** Wen-Bin Liu, Wen-Xuan Pei, Zi-Ming Shao, Jia-Xin Nie, Wei Cao, Chun-Cai Yan

**Affiliations:** Tianjin Key Laboratory of Conservation and Utilization of Animal Diversity, Tianjin Normal University, Tianjin 300387, China; skylwb@tjnu.edu.cn (W.-B.L.); 2310170021@stu.tjnu.edu.cn (W.-X.P.); 2410170020@stu.tjnu.edu.cn (Z.-M.S.); 2410170017@stu.tjnu.edu.cn (J.-X.N.); caowei@stu.tjnu.edu.cn (W.C.)

**Keywords:** chironomids, toxicology, bibliometric analysis, heavy metals, pesticides

## Abstract

This study examines how water pollution harms chironomids—small aquatic insects that help scientists track ecosystem health. By analyzing over 1400 research papers, we found that heavy metals, pesticides, and microplastics are the most studied pollutants, with *Chironomus riparius* being the most common species tested. Research in this field has grown since 1998, showing rising global concern about water pollution. However, more studies are needed at the molecular level to fully understand how these toxins affect insects. This research can help guide better pollution control policies and protect water ecosystems, benefiting both the health of living beings in nature and humans by providing timely biological monitoring information.

## 1. Introduction

The Chironomidae family belongs to the Diptera order and is an important insect species in aquatic ecosystems, widely distributed around the world [1]. In benthic communities, the larvae of the Chironomidae family are the most common species of sedimentary animals in freshwater ecosystems, serving as the primary food source for fish and invertebrates and occupying an important ecological position in freshwater ecosystems [2]. Due to its wide distribution, short lifespan, easy identification of different stages, and high sensitivity to toxins, Chironomidae are widely used as indicator species for environmental pollutants in aquatic ecosystems [3]. Chironomids are used in toxicological experiments and are often employed to detect biological indicators after being treated with chemicals, including population changes (mortality, growth rate, behavior, and reproduction) [4,5], the mouthpart deformities of the larvae [6], and the giant polytene chromosomes from the salivary gland cells [7]. With the continuous acceleration of industrialization, the problem of water pollution is becoming increasingly serious; specifically, the threat of chemical pollutants such as heavy metals and pesticides to aquatic organisms is becoming increasingly prominent [8,9,10,11].

Environmental toxicology studies the impact of environmental toxins on the health of living organisms and the environment [12]. The pollution of water environments mainly includes heavy metals, pesticides and related materials, organic compounds, and sediment. In addition, physical factors such as extreme temperatures and radiation, as well as other biological influences, can cause serious environmental problems. Since the 20th century, the heavy metal content in water environments has rapidly increased mainly due to human activities, including metal mining and smelting, waste leaching, and agricultural use [13,14,15,16,17,18,19]. Multiple heavy metals can accumulate in different organs of humans and cause serious diseases [20]. Pesticides are a type of toxic substance intentionally released by humans to purposefully kill certain organisms, such as weeds, insects, fungi, or rodents [21]. Pesticides are often used in public health activities to control the spread of diseases or prevent the growth of harmful organisms in daily necessities [22]. However, many people may be exposed to certain concentrations of pesticides, which can have adverse effects on their health, especially on children [23,24]. The organic compounds with a high toxicity in aquatic environments are mainly leachable or soluble chemical products, and the hazards of microplastics have been discovered in recent years [25]. Plastic products are released into the air from various sources [26] and cause freshwater pollution [27,28,29]. The accumulation of these microplastics in the human body can also lead to diseases [30,31,32,33,34,35]. In water bodies, various toxic substances can deposit in sediments, causing diverse and complex pollution [36,37,38]. Due to the cocktail effect and synergistic effect of chemical substances, the toxicity of the mixture is also of concern to the public and regulatory agencies [39,40,41]. Physical conditions also have different impacts on aquatic ecology, including salinity stress, temperature, radiation [42,43,44], etc.

Compared to terrestrial organisms, aquatic organisms are more sensitive to exposure and toxicity, and can be used to obtain more accurate data [45]. Therefore, aquatic organisms are used for toxicological research projects [46,47,48]. Among them, as a suitable research species, the chironomid has been extensively studied [49,50,51,52,53]. In the USA, the US Environmental Protection Agency (USEPA) and Organization for Economic Cooperation and Development (OECD) have certified Chironomidae animals as recommended species for testing environmental pollutants in freshwater invertebrates [54,55]. With the increasing importance of water environmental pollution issues and the continuous development of new technologies and molecular-level technologies, it is important to statistically sort out the previous toxicological research on chironomids, focusing on pollutants, research species, and research levels.

In recent years, bibliometrics has gradually become an important analytical tool for scientific research [56]. Bibliometrics can reveal key issues such as disciplinary development trends, research hotspots, and academic influence through the quantitative analysis of a large number of studies [57]. Our study used traditional bibliometric methods and the *GLM 4 Plus* model from *ChatGLM* from Zhipu AI to understand and analyze the toxicological research on chironomids. Based on 1465 articles (Appendix A), we collected bibliometric data and captured the research species, toxicological methods, and research directions in the articles to review the past research progress and provide more directions for future research.

## 2. Materials and Methods

### 2.1. Data Sources and Search Strategies

We searched and obtained relevant data from Web of Science (core collection) and PubMed. Three words related to chironomid were selected to exact terms related to environmental toxicology, so as to comprehensively collect the literature as much as possible. The search formula was conducted as follows: TS = ((“Chironomidae” AND “Toxicology”) OR (“Chironomidae” AND “environmental toxicology”) OR (“Chironomidae” AND “xenobiotic response”) OR (“Chironomidae” AND “xenobiotics”)OR (“Chironomidae” AND “toxic resistance”) OR (“Chironomidae” AND “pollution”) OR (“Chironomidae” AND “environmental contamination”) OR (“Chironomidae” AND “heavy metal”) OR (“Chironomidae” AND “environment”)OR (“Chironomidae” AND “exposure”)OR (“*Chironomus*” AND “Toxicology”) OR (“*Chironomus*” AND “environmental toxicology”) OR (“*Chironomus*” AND “xenobiotic response”) OR (“*Chironomus*” AND “xenobiotics”)OR (“*Chironomus*” AND “toxic resistance”) OR (“*Chironomus*” AND “pollution”) OR (“*Chironomus*” AND “environmental contamination”) OR (“*Chironomus*” AND “heavy metal”) OR (“*Chironomus*” AND “environment”)OR (“*Chironomus*” AND “exposure”)OR (“Chironomid” AND “Toxicology”) OR (“Chironomid” AND “environmental toxicology”) OR (“Chironomid” AND “xenobiotic response”) OR (“Chironomid” AND “xenobiotics”)OR (“Chironomid” AND “toxic resistance”) OR (“Chironomid” AND “pollution”) OR (“Chironomid” AND “environmental contamination “) OR (“Chironomid” AND “heavy metal”) OR (“Chironomid” AND “environment”)OR (“Chironomid” AND “exposure”)). It should be particularly pointed out that there are differences among the terms chironomid, Chironomidae, and *Chironomus*. “Chironomid is a broader term, referring to all the species of Chironomidae, while Chironomidae is the family name and *Chironomus* is the genus name, referring to a genus within Chironomidae.” In order to obtain sufficient articles, the search formula we constructed contains all three words. The main literature type we studied is limited to research articles, and the language used is limited to English. The search result export date was set as 31 December 2024. By comparing the article information obtained from the Web of Science (Core Collection) and PubMed, duplicate articles were screened. In the process of analysis, these articles were regarded as articles from the Web of Science to avoid the excessive counting and analysis of articles. Figure 1 shows the search and filtering strategies of publications.

### 2.2. Data Analysis

For the bibliometric analyses, Microsoft Excel 2020, VOS viewer 1.6.20, CiteSpace 6.3. R1, Scimago Graphica 1.0.46, Hiplot (http://plot.com.cn, accessed on 13 January 2025), Bioinformatics (http://www.Bioinformatics.com.cn, accessed on 15 January 2025), and ChatGLM (http://chatglm.cn, accessed on 5 January 2025) were used. These programs have practical aspects for different analyses.

In this study, VOSviewer 1.6.20 and CiteSpace 6.3.R1 were used to visualize the research results and performance of authors, institutions, countries, and journals in the literature. SCImago Graphica 1.0.46 was used to draw collaborative networks between publishing countries and authors, aiming to reveal development trends. The Hiplot and bioinformatics platform were used to plot the data analyzed in the enrichment analysis. In particular, ChatGLM’s GLM 4 Plus model was used in this study to capture and analyze relevant content. GLM 4 plus model was used to read and analyze the full text of the studies, and capture the research species, research methods, and whether the articles meet the classification as chironomid toxicology research. It is notable that there are conveniences in the analysis using the GLM 4 plus model, but there are also limitations. On the one hand, using large language models can quickly browse a large amount of textual information and capture key information based on preset questions; on the other hand, large language models may misunderstand the captured fragments as key information, even if there are obvious errors in this information in the main text. Therefore, by manually comparing the literature abstracts with the capture results of the GLM 4 plus model, the articles and key information that meet the requirements are selected.

## 3. Results

### 3.1. Publication Patterns

Articles related to ‘Toxicological study by using chironomid’ have been searched in the Web of Science Core Collection (WoSCC) and PubMed. We retrieved 2079 articles from WoSCC and 2168 from PubMed, and 967 duplicate publications were found in both databases. After excluding duplicates, the total number of unique publications was 3280 (Figure 2).

The content was screened to identify toxicology articles using chironomid larvae as experimental materials. After screening, 1815 unrelated articles were excluded. Finally, our study included a total of 1465 publications (Figure 3, Appendix A). The earliest publication was “Studies on Embryonic Determination of the Harlequin-Fly, *Chironomus dorsalis*. II. Effects of Partial Irradiation of The Egg by Ultraviolet Light.”, published in 1964 by YAJIMA, H in *Journal of Embryology and Experimental Morphology*. This study exposed *Chironomus dorsalis* eggs to ultraviolet radiation and described the deformities caused by ultraviolet rays. It is a toxicological article that investigates the impact of physical factors on the morphological changes of species.

The 1465 publications analyzed in this study were authored by 4472 authors from 1330 organizations in 68 countries. Figure 3 shows the overall growth trend of annual and cumulative publications from 1964 to 2024. Since 1998, the number of publications on chironomid toxicology research has been accelerating. In 1999, the number of annual papers entered the triple digits for the first time, reaching 115. In 2009, this number significantly increased to 520 papers, and in 2018, the number of publications exceeded 1000 for the first time. This indicates that there has been significant expansion in the field of chironomid toxicology research in recent years, highlighting the prospects for future development.

### 3.2. Analysis of Journals

A total of 1465 articles were distributed across 229 journals, revealing a notably imbalanced publication distribution. Approximately 65% of journals published fewer than fifteen articles each, while the ten remaining journals each contributed over twenty publications (detailed in Table 1). The three most productive journals in chironomid toxicological research were *Environmental Toxicology and Chemistry* (183 articles), *Science of The Total Environment* (127 articles), and *Ecotoxicology and Environmental Safety* (111 articles).

Impact Factors (IFs) and journal quartiles were obtained from the 2023 Journal Citation Reports (Web of Science). As a key metric of academic influence, the IF reflects the average number of citations received by a journal’s articles over the preceding two-year period. Among the top fifteen most productive journals in this field (listed in Table 1), the three with the highest IF values were *Journal of Hazardous Materials* (IF 12.2), *Water Research* (IF 11.5), and *Environmental Science & Technology* (IF 10.9). These journals, predominantly classified as Q1, demonstrate concentrated quality distribution and underscore their scholarly authority within the discipline.

The Hirsch index (H index), initially designed to measure individual researchers’ academic impact, is now widely used in bibliometrics to evaluate the influence of journals, institutions, and countries. In this analysis, *Environmental Toxicology and Chemistry* ranked highest with an H index of 41, trailed by *Science of The Total Environment* and *Environmental Pollution*, both at 33.

To further assess publication quality, journals were analyzed for Citation Per Paper (CPP) values. The top three journals by CPP were *Environmental Science & Technology* (38.31), *Chemosphere* (32.0), and *Environmental Pollution* (31.74), highlighting their significant academic resonance within the field.

### 3.3. Countries’ and Authors’ Research Performances and Cooperation

Country contributions were evaluated based on author affiliations listed in published articles, with research performance measured through total publications, independent articles, and collaborative works. As shown in Table 2, the top 15 countries—each contributing at least 40 articles—are ranked by total output. The United States led with 331 publications, followed by China (154) and Canada (153). The annual publication growth analysis revealed the U.S. as the fastest-growing contributor (Figure 4, Appendix A). In terms of Citation Per Paper (CPP), Republic of Korea ranked first (31.55), followed by Belgium (31.51) and Spain (31.14), while the United States placed fourth (29.06). The U.S. also dominated the H index rankings with a score of 57, trailed by Canada, Germany, and Spain, each with 37.

To visualize global collaboration patterns in chironomid toxicological research, we graphically mapped international cooperation networks. Figure 5 illustrates the global scope and density of these partnerships within the field. The USA leads in the number of collaborative articles, but in terms of collaboration intensity, it appears more in European countries, with Germany, Portugal, and Spain as the centers of collaboration.

Table 3 ranks the 14 most prolific authors in chironomid toxicological research by publication count. The most numerous are Soares, Amadeu M.V.M and Pestana, Joao L. T. (Universidade de Aveiro, Portugal) with 45 and 32 publications, followed by You, Jing (Jinan University, China) with 29 publications. Regarding citation impact, Morrillo, Gloria (Universidad Nacional de Educación a Distancia, Spain) achieved the highest Citation Per Paper (CPP) value of 47.75, reflecting the exceptional recognition of their work. Lydy, Michael J. (Southern Illinois University, USA) and Choi, Jinhee (University of Seoul, Republic of Korea) ranked closely behind, with CPPs of 45.68 and 43.68, respectively. Collaboration patterns among leading authors are illustrated in Figure 6, which maps the cooperative network of the top 24 researchers based on collaboration intensity.

### 3.4. The Most Cited Articles and High-Impact Articles in 1465 Publications

The citation frequency of an article can reflect the research focus and trends in a specific field. Table 4 shows 15 cited articles out of 1465 publications with at least 140 citations. The analysis of the top fifteen most frequently cited articles shows that there are a maximum of four articles from the United States, with the remaining two articles from South Africa and Australia. In addition, there are four articles from international cooperation: South Africa and Malaysia, Germany and Norway, USA and China, and Netherlands and Bolivia.

As mentioned earlier, the publication with the most citations was “Sinks and sources: Assessing microplastic abundance in river sediment and deposit feeders in an Austral temperate urban river system”, published by Rhodes University in *Science of the Total Environment*. Since its publication in 2018, it has been cited 343 times. The second most cited publication was “Feeding type and development drive the ingestion of microplastics by freshwater invertebrates”, published by Goethe University Frankfurt in *Scientific Reports*. Since its publication in 2017, it has been cited 300 times. Both articles evaluated microplastic pollution using Chironomidae. These articles separately tracked the dynamic changes in microplastic pollution in the Bloukrans River system in South Africa during different seasons and the uptake rate of *Chironomus riparius* exposed to microplastics of different sizes and concentrations. Both focus on the uptake of microplastics by species and explore the impact of environmental factors and species characteristics on microplastic uptake.

However, there are some articles that are equally important, even if they are not cited as frequently. “Biofragmentation of Polystyrene Microplastics: A Silent Process Performed by *Chironomus sancticaroli* Larvae” was published by Universidade de Sao Paulo in *Environmental Science & Technology*, the JCR Category Quartile of which is Q1, with a five-year impact factor of 11.7, but it has only been cited 10 times. This article studies the ability of *Chironomus sancticaroli* to promote the fragmentation of Polystyrene microspheres and the toxic effects of exposure to this polymer. It is the first study to report the ability of chironomid to promote the biofragmentation of microplastics. However, since this article was published on 1 March 2024, it may not have received much attention yet.

### 3.5. Analysis of Keywords Network

Keywords embody the core and essence of a paper, and co-occurrence analysis of keywords can identify research hotspots in specific scientific fields. 86 keywords mentioned 9 or more times in 1465 publications were selected to create a keyword co-occurrence network (Figure 7). This keyword network illustrates the emergence trend and connection intensity of these keywords of publications related to chironomid toxicology research.

The central terms in the keyword network are *Chironomus riparius*, Chironomidae, and ecotoxicology. According to the connections between terms, keywords can be divided into four categories: heavy metals (such as cadmium, copper, zinc, etc.), pesticide (such as Chlorpyrifos, Pyrethroids, Atrazine, etc.), insect groups (such as *Chironomus riparius*, *Chironomus riparius, Propsilocerus akamusi*, etc.), and stresses types (such as sediment toxicity, oxidative stress, temperature, etc.).These keywords highlight the impact of environmental pollutants and different stress factors on Chironomidae. Therefore, we systematically analyzed 1465 studies on the effects of heavy metals, pesticides, and environmental stress on Chironomidae insects, exploring the toxic mechanisms and physiological responses of different pollutants on insects to further understand the research trends and current status in this field.

## 4. Discussion

### 4.1. Analysis of Target Species and Environmental Stressors

In the study of stress on chironomid species, important trends can be seen from the number and distribution of studies on different stress materials. Figure 8 illustrates the number of publications per species and the composition of treatment methods applied to each species (Appendix A).

*Chironomus riparius,* as a common model species, accounts for 38.05% of research and plays a significant role in ecological toxicology and environmental pollution research [58,59,60,61,62,63,64]. Following closely behind are Chironomidae, accounting for 14.08% [65,66,67,68,69,70,71], and *Chironomus dilutes* [72,73,74,75,76] and *Chironomus tentans* [77,78,79,80], accounting for 9.14% and 8.76%, respectively, which are widely used in ecological research. Among the top ten species studied, *Chironomus* spp. have the largest total amount of research. Chironomus spp. is one of the most important functional groups in the soft-bottom communities of freshwater ecosystems [81]. Furthermore, *Chironomus riparius*, *Chironomus dilutes*, and *Chironomus tentans* have been certified by the US Environmental Protection Agency and the OECD as recommended species for toxicology research, which is also the reason for the greater number of publications [54,55].

We categorize the adverse factors on chironomids into heavy metals, inorganic substances (non-heavy metals), pesticides, organic substances (non-pesticides), sediments, biological effects, and physical treatments. Among them, heavy metal and organic matter (non-pesticide) pollution are the most studied stress materials, with heavy metal-related research accounting for 27.16% and organic matter (non-pesticide) research accounting for 26.60%. This indicates that the ecological impact of these two types of pollutants on chironomid species is currently a key research area, possibly due to their widespread presence in aquatic ecosystems and significant impact on the growth and development of aquatic organisms. Heavy metals such as Cu [68,69,82], Hg [83,84], Cd [61,85,86], Pb [87,88], and Zn [89,90] exhibit persistence and bioaccumulation in aquatic environments, while non-pesticide organic compounds include various compounds [59,63,65,74,91]. Among non-pesticide inorganic pollutants, microplastics are becoming a research hotspot. Over the past two decades, hundreds of papers have specifically studied the accumulation of microplastics in the environment, indicating that microplastics have polluted a variety of environments worldwide [92]. The international legally binding instrument on plastic adopted by the United Nations Environment Programme (UNEP) regards microplastics, plastic materials and products, and plastic-related chemicals as key aspects of plastic pollution [93]. Their potential threat to ecosystems is gradually being recognized by the academic community. Applying pesticides is also a common treatment method, and common pesticides such as thiamethoxam (TMX) [94,95,96], fipronil (FIP) [97,98], 2,4-D [99,100], and deltamethrin [101,102,103] are soluble in water, which may have adverse effects on the water body. Excluding heavy metals, the inorganic compounds that are mentioned more frequently include nitrogen-containing compounds [66,71,104] and selenium-containing compounds [105,106,107].

There is limited research on biological effects and physical treatments, and for biological effects, there are only two types of organisms: *Vibrio cholerae* [108,109,110,111] and *Bacillus thuringiensis* israelensis (Bti) [62,112,113,114]. *Vibrio cholerae* is an aquatic bacterium that can cause large-scale human infectious disease cholera, while Bti is widely used as a microbial insecticide. Vibrio cholerae is a common host of chironomids, and Btis can act as an insecticide and play a role similar to that of organic pesticides. Therefore, the research mainly focuses on these two aspects, while there are relatively few other studies. The common treatment method in physical processing is to change the temperature [115,116,117]. Compared to other studies, *Belgica antarctic* has appeared more frequently in research on temperature [118,119,120], possibly due to the fact that *Belgica antarctica* is the only insect species living on the Antarctic Peninsula and its nearby islands. Another common physical treatment method is dehydration, and *Polypedilum vanderplanki* has received more attention due to its strong dehydration ability, living in arid Africa [121,122,123]. Overall, researchers tend to focus on studying the unique chironomid in extreme environments in physical studies. One possible reason for the relatively small amount of research on physical factors is that more toxicology researchers mainly focus on the reactions between chemical substances and organisms, and thus do not consider physical factors as part of toxicology research.

### 4.2. Species Response Analysis

The level of attention to changes in chironomids varies in different studies. Figure 9 shows chironomid toxicology at different times. The quantity and composition of publications are shown at the research level (Appendix A).

The apparent level is in a relatively advantageous position in each period, followed by the molecular level, and the research on enzyme-level changes is the least. However, with the increase of years, various studies have shown a certain growth trend, with the enzyme level showing the fastest growth rate. The apparent level is widely used in research because its indicators mainly include two parts: population changes and individual structural changes. The population changes led by mortality rate are being used as indicators by more researchers [124,125,126,127], which is easy to statistically analyze; individual structural changes such as head capsule distortion are also significant changes [95,128], which can intuitively reflect the toxicological stress situation. At the molecular level, research focuses more on chromosomal aberrations and Hsp70 [115,129,130,131,132]; enzyme levels tend to be more associated with changes in AChE, SOD, GST, and CAT, which are often related to detoxification functions [133,134,135,136]. Compared with the apparent level, the enzyme level can better understand the changes in species’ response to stress at the microscopic level; however, compared with the molecular research level, it is easier to obtain samples with enzyme levels. Moreover, most of the research indicators of enzyme levels have mature detection methods and standards, allowing researchers to conveniently measure and compare, thus making it easier to obtain reliable data and conclusions. This has promoted the rapid progress of enzyme-level research.

## 5. Conclusions

Chironomid larvae have become an ideal model organism for toxicology research in freshwater ecosystems due to their wide distribution, short life cycle, strong sensitivity, and ease of laboratory cultivation [137]. The main research subjects include *Chironomus riparius* (38.05% of the study), *Chironomus*
*dilutus* (9.14%), and *Chironomus tentans* (8.76%), which have been certified as standardized test species by the US Environmental Protection Agency (USEPA) and the Organization for Economic Cooperation and Development (OECD). Most research areas focus on the toxic effects of heavy metals (such as cadmium, copper, and zinc, accounting for 27.16%), non-pesticide organic pollutants (such as microplastics and PFAS, accounting for 26.60%), and pesticides (such as thiamethoxam and fipronil) on chironomids, while also paying attention to the impact of extreme environments (such as temperature and dehydration) on their physiological and molecular responses.

The current research still faces some problems; for example, some studies have shown that physiological changes are not as sensitive as molecular changes [138], which may require further research to focus on molecular level changes rather than staying at the individual level, and further research can explore how environmental pollutants affect the toxic mechanisms of chironomids at the molecular level. Future chironomid toxicological experimental research can not only remain at the level of detecting toxicological indicators of environmental pollutants on chironomids, but also utilize molecular biology techniques, such as the CRISPR gene-editing technique, etc. This technique is used to actively explore the interaction mechanism of biological macromolecules of species in the detoxification response. Relevant regulatory organizations or institutions (for example, USEPA or OECD) should also actively update more comprehensive norms in accordance with the latest research progress.

## Figures and Tables

**Figure 1 insects-16-00639-f001:**
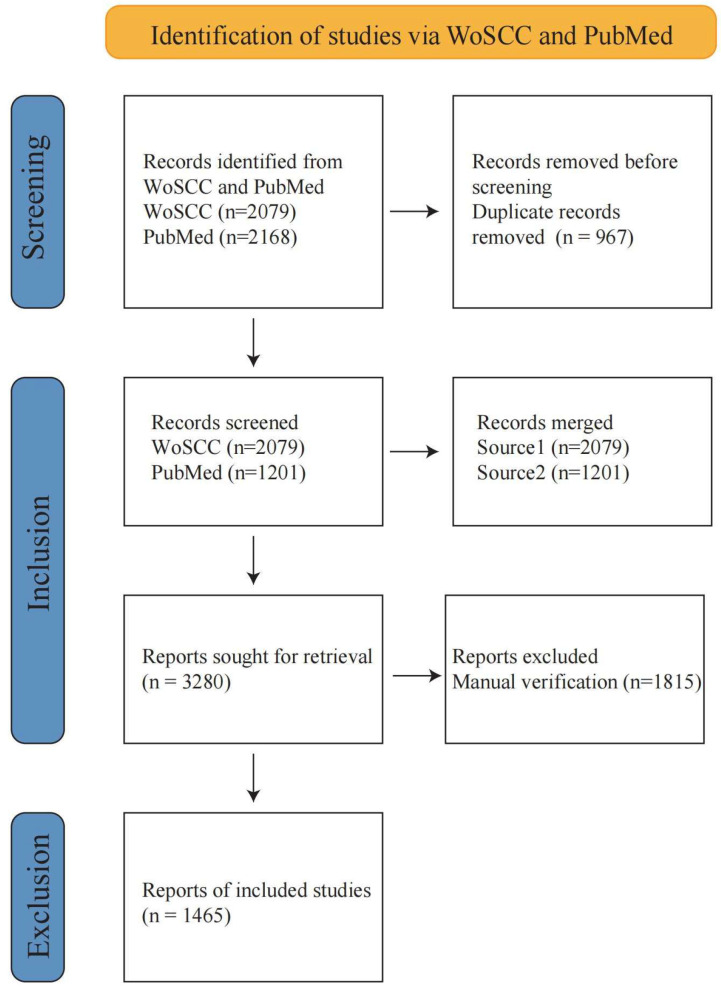
The flowchart for screening articles from WosCC and PubMed. In the inclusion step stage, articles from two sources are marked and downloaded. The original texts are uploaded to the *GLM 4 plus* model to extract information. The summaries and the results returned by the *GLM 4 plus* model are read manually to make the final screening.

**Figure 2 insects-16-00639-f002:**
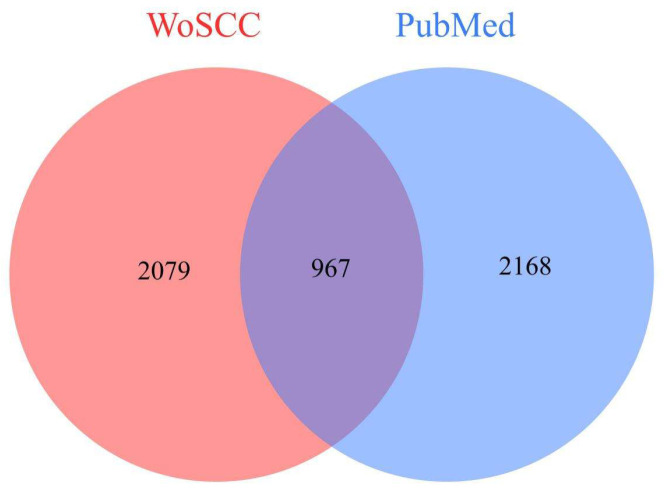
Venn diagram of total articles retrieved from WoSCC and PubMed databases. Venn diagram of total articles retrieved from WoSCC and PubMed databases. The red section represents 2079 articles from WoSCC, the blue section represents 2168 articles from PubMed, and the overlapping area represents the total of 967 duplicated articles.

**Figure 3 insects-16-00639-f003:**
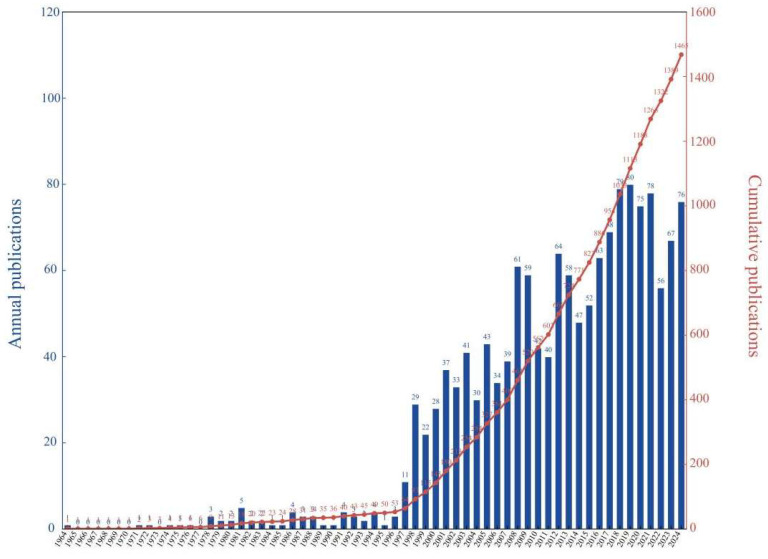
Annual publication quantity and growth trend. The bar chart represents the number of newly issued publications each year, while the line chart represents the total cumulative number of publications issued up to the current year.

**Figure 4 insects-16-00639-f004:**
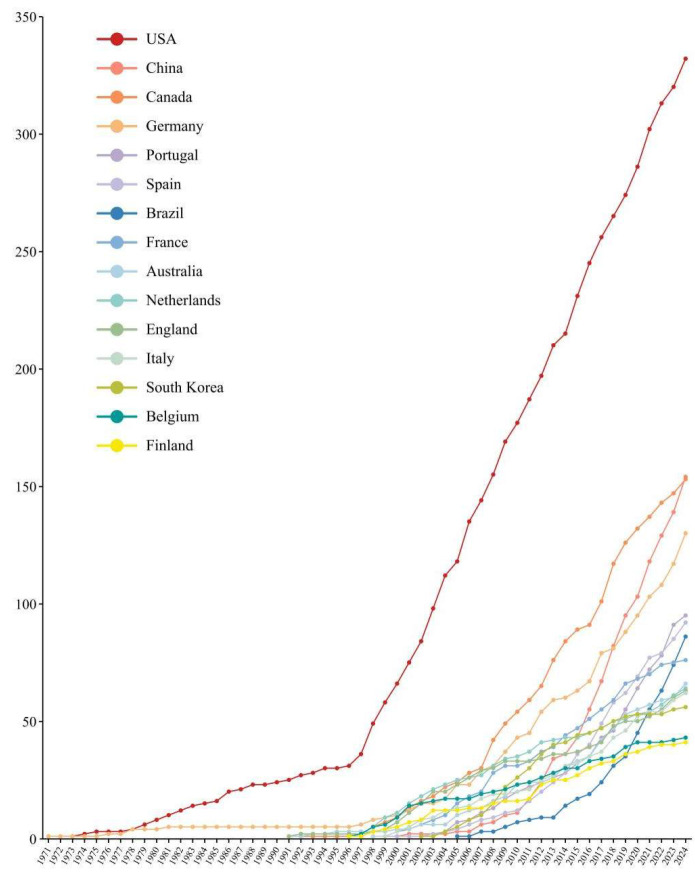
Annual growth of publications in the top 15 countries. Different colored lines represent the total number of publications published by different countries as of the current year.

**Figure 5 insects-16-00639-f005:**
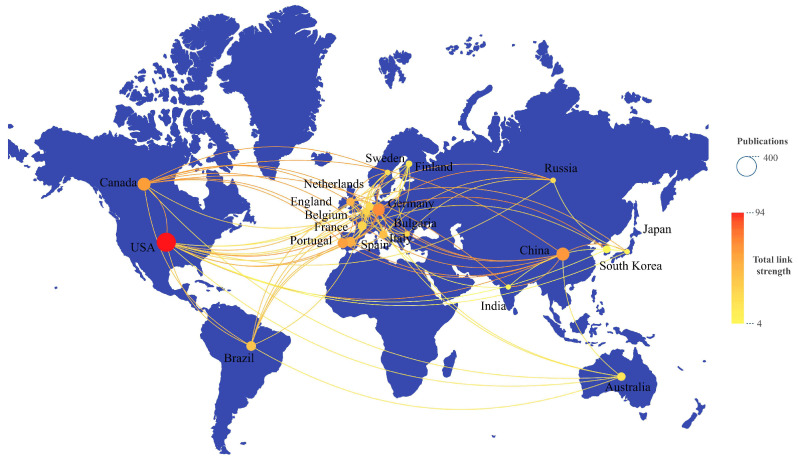
Geographic extent of collaborations for research in toxicological study on chironomids. The size of the label indicates the number of publications; the more countries that are involved, the redder the color of the label.

**Figure 6 insects-16-00639-f006:**
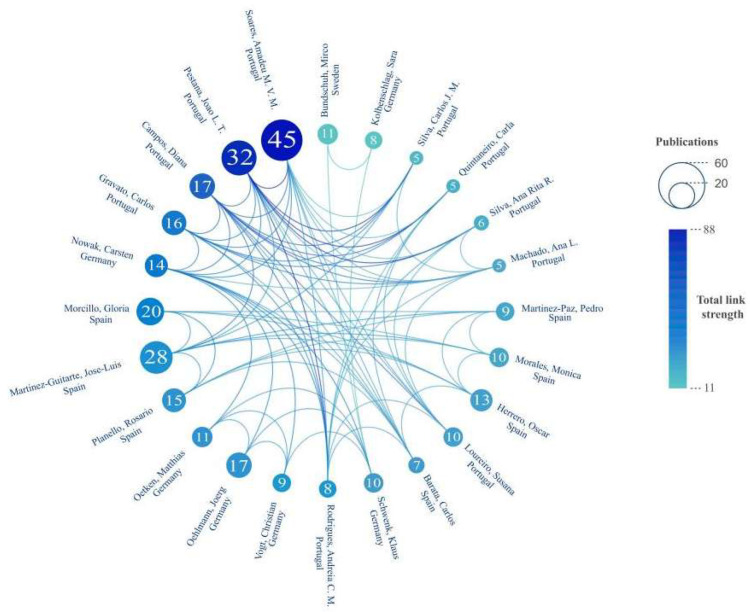
The collaborative network of the top 24 authors with the highest collaboration intensity. The lines represent the author’s collaborative relationship, with stronger lines indicating greater collaboration strength. The larger the circle size, the more collaborative publications there are. The darker the color, the greater the total link strength.

**Figure 7 insects-16-00639-f007:**
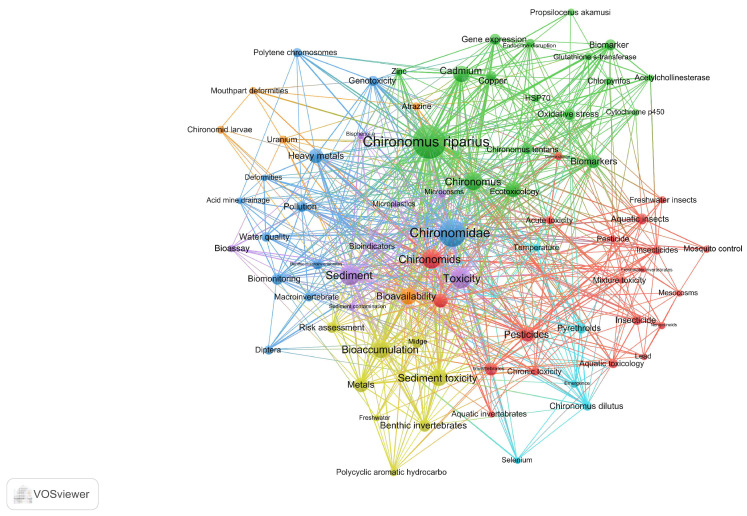
The co-occurrence map of keywords. The larger the circular node, the higher the frequency of keyword occurrence, and the more it can represent the hotspots in the field. The connecting lines between nodes represent the strength of the association. The thicker the connecting line, the higher the frequency of these two keywords appearing in the same literature.

**Figure 8 insects-16-00639-f008:**
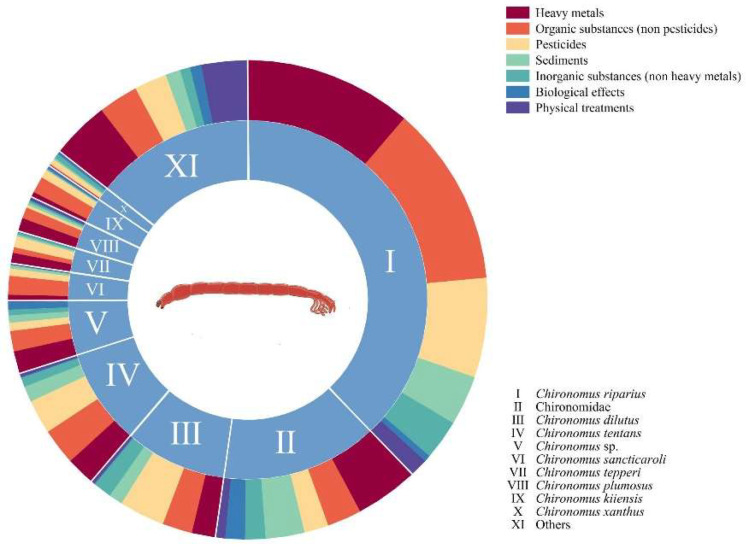
Species and treatment methods. The inner circle represents different varieties used in the research, with I to XI corresponding to the species groups marked in the legend at the lower right corner; the different colors of the outer circle represent different treatment methods, which are explained in the legend at the upper right corner. The proportion of each area reflects the number of corresponding studies; the larger the proportion area, the more the number of studies.

**Figure 9 insects-16-00639-f009:**
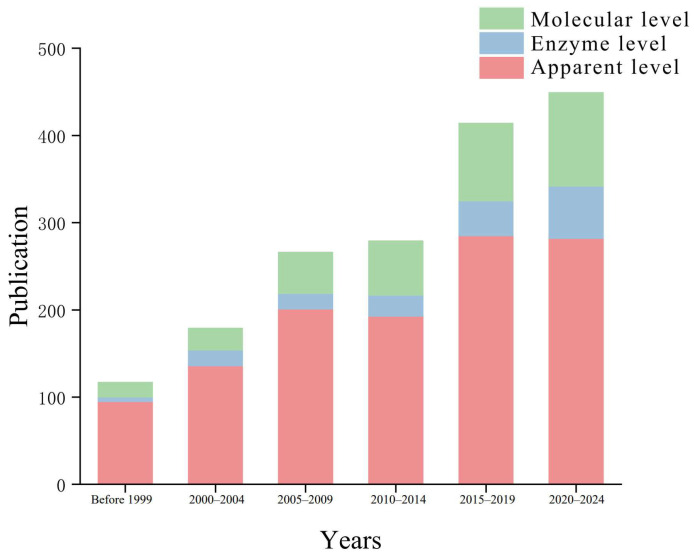
Changes in research direction over the years. Starting from 2000, every five years are counted as a period, and the entire period before 1999 is counted as a period. The height of the bar chart reflects the number of publications over a certain period of time. The proportions of red, blue, and green parts represent the number of articles related to apparent, enzymatic, and molecular levels, respectively.

**Table 1 insects-16-00639-t001:** The top 15 journals with the most publications, with publisher, number of publications, citations, CPP, h-index, IF, and JIF quarters.

Rank	Journal	Publisher	Publications	Citations	CPP	h-Index	IF	JIF Quartile
1	*Environmental Toxicology and Chemistry*	WILEY	183	5558	30.37	41	3.6	Q2
2	*Science of The Total Environment*	ELSEVIER	127	3379	26.61	33	8.2	Q1
3	*Ecotoxicology and Environmental Safety*	ACADEMIC PRESS INC ELSEVIER SCIENCE	111	2631	23.70	29	6.2	Q1
4	*Environmental Pollution*	ELSEVIER SCI LTD	99	3142	31.74	33	7.6	Q1
5	*Chemosphere*	PERGAMON-ELSEVIER SCIENCE LTD	79	2528	32.00	32	8.1	Q1
6	*Archives of Environmental Contamination And Toxicology*	SPRINGER	68	1975	29.04	25	3.7	Q2
7	*Aquatic Toxicology*	ELSEVIER	58	1632	28.14	25	4.1	Q1
8	*Environmental Science & Technology*	AMER CHEMICAL SOC	51	1954	38.31	25	10.9	Q1
9	*Bulletin of Environmental Contamination and Toxicology*	SPRINGER	42	690	16.43	16	2.7	Q3
10	*Environmental Science and Pollution Research*	SPRINGER HEIDELBERG	35	520	14.86	14	5.8	Q1
11	*Ecotoxicology*	SPRINGER	32	628	19.63	14	2.5	Q3
12	*Environmental Monitoring and Assessment*	SPRINGER	21	279	13.29	12	2.9	Q3
13	*Journal of Hazardous Materials*	ELSEVIER	19	333	17.53	10	12.2	Q1
14	*Water Research*	PERGAMON-ELSEVIER SCIENCE LTD	18	464	25.78	13	11.5	Q1
15	*Water Air and Soil Pollution*	SPRINGER INT PUBL AG	18	260	14.44	10	3.8	Q3

**Table 2 insects-16-00639-t002:** The top 15 countries and regions with the most publications, with the number of publications, citations, CPP, and H index.

Rank	Country	Publications	Citations	CPP	H Index
1	USA	331	9618	29.06	57
2	China	154	3009	19.54	31
3	Canada	153	4045	26.44	37
4	Germany	130	3463	26.64	37
5	Portugal	95	1875	19.74	27
6	Spain	92	2865	31.14	37
7	Brazil	86	1062	12.35	18
8	France	76	1815	23.88	28
9	Australia	66	1493	22.62	22
10	Netherlands	64	1811	28.30	27
11	England	63	1582	25.11	27
12	Italy	62	1331	21.47	26
13	Republic of Korea	56	1767	31.55	26
14	Belgium	43	1355	31.51	26
15	Finland	40	805	20.13	19

**Table 3 insects-16-00639-t003:** The top 14 authors with publications greater than 15, with countries, institution, number of publications, citations, and CPP.

Rank	Authors	Country	Institution	Publications	Citations	CPP
1	Soares, Amadeu M. V. M	Portugal	Universidade de Aveiro	45	1076	23.91
2	Pestana, Joao L. T.	Portugal	Universidade de Aveiro	32	879	27.47
3	You, Jing	China	Jinan University	29	761	26.24
4	Choi, Jinhee	Republic of Korea	University of Seoul	28	1223	43.68
5	Martinez-Guitarte, Jose-Luis	Spain	Universidad Nacional de Educacion a Distancia	28	1001	35.75
6	Liber, Karsten	Canada	University of Saskatchewan	28	843	30.11
7	Pettigrove, Vincent	Australia	University of Melbourne	25	477	19.08
8	Lydy, Michael J.	USA	Southern Illinois University	22	1005	45.68
9	Morcillo, Gloria	Spain	Universidad Nacional de Educacion a Distancia	20	955	47.75
10	Campos, Diana	Portugal	Universidade de Aveiro	17	393	23.12
11	Li, Huizhen	China	Jinan University	17	320	18.82
12	Oehlmann, Joerg	Germany	Goethe University Frankfurt	17	499	29.35
13	Gravato, Carlos	Portugal	Universidade de Aveiro	16	526	32.88
14	Planello, Rosario	Spain	Universidad Nacional de Educacion a Distancia	15	549	36.60

**Table 4 insects-16-00639-t004:** The top 15 most cited articles include first address country, other participating countries, first address participating institutions, and citations.

Rank	Publication	First Address Country	Other Participating Countries	First Address Participating Institutions	Citations
1	Sinks and sources: Assessing microplastic abundance in river sediment and deposit feeders in an Austral temperate urban river system	South Africa	Malaysia	Rhodes University	343
2	Feeding type and development drive the ingestion of microplastics by freshwater invertebrates	Germany	Norway	Goethe University Frankfurt	300
3	Distribution and toxicity of sediment-associated pesticides in agriculture-dominated water bodies of California’s Central Valley	USA	N/A	University of California System	254
4	Environmentally relevant concentrations of polyethylene microplastics negatively impact the survival, growth and emergence of sediment-dwelling invertebrates	Australia	N/A	Griffith University	214
5	Waterborne and sediment toxicity of fluoxetine to select organisms	USA	N/A	Baylor University	188
6	Fluctuating asymmetry of invertebrate populations as a biological indicator of environmental quality.	Australia	N/A	CSIRO Division of Entomology	181
7	Anthropogenic impacts on the distribution and biodiversity of benthic macroinvertebrates and water quality of the Langat River, Peninsular Malaysia	Malaysia	N/A	Universiti Putra Malaysia	176
8	Partitioning, bioavailability, and toxicity of the pyrethroid insecticide cypermethrin in sediments	England	N/A	Syngenta	174
9	Mechanism allowing an insect to survive complete dehydration and extreme temperatures	Japan	N/A	National Institute of Agrobiological Sciences-Japan	167
10	Impact of atrazine on organophosphate insecticide toxicity	USA	N/A	Wichita State University	159
11	Expression of heat shock protein and hemoglobin genes in *Chironomus tentans* (Diptera, Chironomidae) larvae exposed to various environmental pollutants: A potential biomarker of freshwater monitoring	Republic of Korea	N/A	University of Seoul	152
12	Effectiveness of a constructed wetland for retention of nonpoint-source pesticide pollution in the Lourens River catchment, South Africa	South Africa	N/A	Stellenbosch University	145
13	Temperature as a toxicity identification evaluation tool for pyrethroid	USA	China	Southern Illinois University System	144
14	Effects of mining activities on heavy metal concentrations in water, sediment, and macroinvertebrates in different reaches of the Pilcomayo River, South America	Netherlands	Bolivia	Radboud University Nijmegen	143
15	Bridging levels of pharmaceuticals in river water with biological community structure in the Llobregat river basin (Northeast Spain)	Spain	N/A	University of Barcelona	143

N/A: Not applicable. There are no other participating countries here.

## Data Availability

The raw data supporting the conclusions of this article will be made available by the authors on request.

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
