# Peer review of "Trends and Emerging Hotspots in Toxicology of Chironomids: A Comprehensive Bibliometric Analysis"

_insects, 2025, doi:10.3390/insects16060639_

Round 1
Reviewer 1 Report
Comments and Suggestions for Authors
General Comments
The manuscript provides a comprehensive and timely bibliometric analysis of research trends in chironomid toxicology. It offers valuable insights into key pollutants, species used, global research contributions, and the evolution of research focus (apparent, enzymatic, molecular). The use of various visualization and bibliometric tools (VOSviewer, CiteSpace, ChatGLM, etc.) strengthens the analysis.
However, there are areas that need clarification, reorganization, and editing to enhance scientific rigor and clarity for international readers.
Major Comments
1. Clarity and Focus of Objectives
The abstract and introduction lack a clearly stated research hypothesis or question. While the purpose is generally stated as bibliometric analysis, specifying a goal like identifying gaps or forecasting future directions would add focus.
Suggestion: Add a sentence to the abstract and introduction clarifying why this analysis is critical now—e.g., increased pollution pressures, need for molecular-level insights, etc.
2. Search Strategy and Inclusion Criteria
The search formula is extensive, but it is not clear if MeSH terms or controlled vocabulary were used in PubMed.
It is also unclear if gray literature, non-English papers, or preprints were excluded.
Suggestion: Provide a PRISMA-style flow diagram showing screening, inclusion, and exclusion steps.
3. Terminology Consistency
There is inconsistency in the use of terms: "chironomid," "Chironomidae," and "Chironomus" are used interchangeably without clarifying their taxonomic hierarchy.
Suggestion: Standardize terminology throughout and clarify early on that "chironomid" is a broader term, while Chironomus riparius is the model organism.
4. Quantitative Bias
The analysis heavily favors frequency metrics (e.g., number of publications) over impact metrics (e.g., breakthrough findings, policy influence).
The overemphasis on quantity might overlook seminal low-frequency papers with high scientific impact.
Suggestion: Include a section on “high-impact outliers” or “landmark studies” that shaped the field despite low volume.
5. Data Reliability and Use of ChatGLM
Use of ChatGLM 4 Plus for literature analysis is interesting but raises concerns:
How reliable is it for full-text semantic analysis?
Was it validated against manual curation?
Suggestion: Discuss the limitations or validation of using AI models for bibliometric screening in the Methods section.
6. Figure Quality and Integration
Figures (especially Figures 6–8) are not always clearly labeled or referenced in the text.
Color codes in circular plots (Figure 7) need legends in the same frame for accessibility.
Suggestion: Revise figure captions to be more explanatory and self-contained.
7. Discussion Depth
While the Discussion summarizes data well, it lacks critical interpretation or hypothesis generation.
For example, why is enzymatic-level research growing fastest? Is this due to ease of assays, funding trends, or technological advances?
Suggestion: Integrate possible causes behind trends, not just observe them.
Minor Comments
Line 15: "benefiting both wildlife and human health" – too general. Consider specifying how (e.g., through biomonitoring).
Line 126: Typo: “judgment of chironomid toxicology” → perhaps “classification as chironomid toxicology”?
Line 215: Spell out author institutions consistently.
Author Response
To Reviewer #1:
Major Comments:
- Clarity and Focus of OGeneral Comments
Comment 1: The abstract and introduction lack a clearly stated research hypothesis or question. While the purpose is generally stated as bibliometric analysis, specifying a goal like identifying gaps or forecasting future directions would add focus.
R 1: We appreciate your feedback and have now added corresponding descriptions in both the abstract and the introduction.
- Search Strategy and Inclusion Criteria
Comment 2: The search formula is extensive, but it is not clear if MeSH terms or controlled vocabulary were used in PubMed. It is also unclear if gray literature, non-English papers, or preprints were excluded.
R 2: Thank you for your valuable feedback. We used the same search formula in both WoSCC and PubMed (as shown in the article), so we did not make any specific adjustments to the terms in PubMed. During the search, when we export from the website search, we choose not to export other literature, including reviews, non-English articles, and other situations. Therefore, only published English articles are available. Meanwhile, we have added the relevant flowcharts here.
- Terminology Consistency
Comment 3: There is inconsistency in the use of terms: "chironomid," "Chironomidae," and "Chironomus" are used interchangeably without clarifying their taxonomic hierarchy.
R 3: Thank you for your suggestion. We have added specifications for chironomid, Chironomidae, and Chironomus at the search formula section. While standardizing the terms, we also explain the reasons for the appearance of these terms in the search formula.
- Quantitative Bias
Comment 4: The analysis heavily favors frequency metrics (e.g., number of publications) over impact metrics (e.g., breakthrough findings, policy influence). The overemphasis on quantity might overlook seminal low-frequency papers with high scientific impact.
R 4: Thank you for your insightful point on this. We have added this section by presenting an example that is highly influential and pioneering but has a low citation rate.
- Data Reliability and Use of ChatGLM
Comment 5: Use of ChatGLM 4 Plus for literature analysis is interesting but raises concerns:
How reliable is it for full-text semantic analysis? Was it validated against manual curation?
R 5: Thank you for pointing this out. We have added a discussion on this aspect in the methods section
- Figure Quality and Integration
Comment 6:Figures (especially Figures 6–8) are not always clearly labeled or referenced in the text. Color codes in circular plots (Figure 7) need legends in the same frame for accessibility.
R 6: Thank you for your careful review and valuable comments. We have revised the description of the picture caption (of course, with the picture inserted in Issue2, it is now Figure7-9) to make the caption more detailed and specific.
- Discussion Depth
Comment 7: While the Discussion summarizes data well, it lacks critical interpretation or hypothesis generation. For example, why is enzymatic-level research growing fastest? Is this due to ease of assays, funding trends, or technological advances?
R 7: Thank you for your valuable suggestions. We added the analysis and explanation of this part, trying to make the discussion more meaningful.
Comment 8: Minor Comments
R 8: Thank you for your detailed and careful review. We have revised the text description based on the suggestions you put forward.
Reviewer 2 Report
Comments and Suggestions for Authors
The authors have addressed succesuflly all of my comments/suggestions and therefore i recommend publication of this work
Author Response
To Reviewer #2:
Comment 1: The authors have addressed succesuflly all of my comments/suggestions and therefore i recommend publication of this work.
R 1: Thank you for carefully and meticulously reviewing the manuscript and making valuable comments. In the latest version of the manuscript, we updated some textual descriptions, added a flowchart for explaining the work, readjusting some sentences in the articles, and improved the persuasiveness and scientific nature of the articles. Thank you again for your recognition and support for our work.